# Efficacy, Safety and Economic Evaluation of Wolbigachul-Tang for Chronic Cough Due to Upper Airway Cough Syndrome (UACS): A Study Protocol for Randomized, Double-Blind, Active-Comparator Controlled, Parallel, Exploratory Clinical Trial

**DOI:** 10.3390/healthcare11202733

**Published:** 2023-10-13

**Authors:** Seong-Cheon Woo, Yee Ran Lyu, Su Won Lee, O-Jin Kwon, Young-Eun Choi, Changsop Yang, Yang Chun Park

**Affiliations:** 1Division of Respiratory Medicine, Department of Internal Medicine, College of Korean Medicine, Daejeon University, Daejeon 35235, Republic of Korea; wsc672@naver.com (S.-C.W.); tndnjs3325@daum.net (S.W.L.); 2KM Science Research Division, Korea Institute of Oriental Medicine, Daejeon 34054, Republic of Korea; ondeoctor2ran@kiom.re.kr (Y.R.L.); cheda1334@kiom.re.kr (O.-J.K.); 3Clinical Research Coordinating Team, Korea Institute of Oriental Medicine, Daejeon 34054, Republic of Korea; wowo9129@kiom.re.kr

**Keywords:** upper airway cough syndrome, chronic cough, Wolbigachul-tang, Korean medicine, herbal medicine

## Abstract

Upper airway cough syndrome (UACS) is a common cause of chronic cough characterized by upper airway symptoms, including nasal discharge and throat discomfort. Empirical treatments for UASC-induced chronic cough, such as first-generation antihistamines, have been used; however, the long-term use of these medicines has adverse effects. Therefore, we evaluate the efficacy, safety, and economic feasibility of Wolbigachul-tang (WBGCT), an herbal medication for UASC-induced chronic cough. This is a randomized, double-blind, active-comparator-controlled, parallel, and exploratory clinical trial. Thirty patients with UASC-induced chronic cough will be recruited and randomly allocated to the WBGCT and control groups in a 1:1 allocation ratio. The investigational medicine will be administered three times per day for 2 weeks (3 g of WBGCT at a time). The primary outcome measure is the cough symptom score measured at screening, before starting the trial, and after 2 and 4 weeks. Secondary outcome measures include the cough visual analog scale, nasal discharge score, questionnaire of clinical symptoms of cough and sputum, Leicester cough questionnaire-Korean version, integrative medicine outcome scale, integrative medicine patient satisfaction scale, and 5-level EuroQol 5-dimensional questionnaire, which will be assessed before starting the trial and after 2 and 4 weeks. This study aims to investigate the efficacy, safety, and economic feasibility of WBGCT in the treatment of chronic cough. Therefore, the results of this trial provide evidence for the application of WBGCT in the treatment of UACS-induced chronic cough.

## 1. Introduction

Cough is one of the most common symptoms of respiratory diseases and is classified as acute, subacute, or chronic, depending on its duration [1]. Among these, the importance of chronic cough lasting longer than 8 weeks has emerged because it has been correlated with decreased quality of life, including physical, mental, and social aspects, as well as complications such as vomiting, nausea, chest pain, urinary incontinence, anxiety, and depression [2,3]. Common causes of chronic cough include upper airway cough syndrome (UACS), cough-variant asthma, and gastroesophageal reflux disease, and of these, UACS is the most common cause of chronic cough [4], accounting for 19–25% of the causes of chronic cough [5,6].

UACS is characterized by a chronic cough with upper airway symptoms, including a sensation of nasal secretions in the throat, nasal stuffiness, rhinorrhea, and throat itching, resulting in physical and psychosocial complications [7,8]. The treatment of patients with UACS-induced cough depends on empirical therapy. When the cause of UACS, such as rhinitis or sinusitis, is obvious, treatment focusing on the cause should be administered. While specific causes may not be identified, medications that are effective in treating UACS-induced cough should be considered [9]. First-generation antihistamines or decongestants are empirically administered to patients with UACS-induced cough. If patients have persistent nasal symptoms, nasal steroids or anticholinergic agents should be administered [10]. However, evidence that antihistamines are effective in the treatment of UACS is insufficient, and long-term use of these medications has adverse effects, including drowsiness, fatigue, memory loss, and decreased concentration [11,12].

Previous studies have demonstrated the therapeutic effects of herbal medications in patients with UACS [13,14]. Wolbigachul-tang (WBGCT) is an herbal formula used to treat respiratory diseases such as asthma and allergic rhinitis [15,16]. It has been reported that WBGCT has antioxidant, anti-inflammatory, and antitussive effects on laryngeally-induced cough [17,18]. The hypothesized mechanism by which WBGCT has therapeutic effects against UACS-induced chronic cough is through the alleviation of upper airway inflammation. However, since the adverse effects of WBGCT including palpitation and insomnia have been reported [19], we will assess the safety of WBGCT, as well as its efficacy. In addition, to explore the possibility of using WBGCT with health insurance coverage, we will evaluate the economic feasibility of WBGCT by comparing it with Socheongryong-tang (SCRT), a health insurance-covered herbal formula used for respiratory symptoms. SCRT has been used for respiratory diseases, such as allergic rhinitis, asthma, and bronchitis [20,21], and has anti-inflammatory and anti-allergic effects that are promoted by inhibiting cytokine release, infiltration of inflammatory cells, and airway remodeling in animal models [22,23]. The safety of SCRT has been demonstrated in previous clinical trials and in vivo studies [24,25]. Considering its efficacy and safety, SCRT was selected as the investigational medicine in this study.

This study aims to investigate the efficacy, safety, and economic evaluation of WBGCT for UACS-induced chronic cough, a feasibility study to demonstrate its possibilities, including the duration of intervention, recruitment of participants, outcome measures, and effect size before this trial.

## 2. Methods

### 2.1. Study Design

This is a randomized, double-blind, active-comparator-controlled, parallel, exploratory clinical trial to evaluate the efficacy, safety, and economic feasibility of WBGCT for UACS-induced chronic cough. Thirty participants will be recruited for this trial and randomly assigned to the WBGCT and control groups in a 1:1 allocation ratio. This trial consists of screening, 2-week treatment, and 2-week observation periods. The medications will be administered three times a day for 2 weeks. The detailed study design is summarized in Figure 1 and Table 1.

### 2.2. Participants

#### 2.2.1. Inclusion/Exclusion Criteria

The inclusion criteria are as follows: (1) men and women aged 19–70 years; (2) patients with symptoms of rhinitis or postnasal drip (rhinorrhea, nasal discharge, cough, or nasal itching); or with gross findings of pebble-like appearance in pharyngeal mucosa; or dripping mucus in the throat or with radiological findings of sinusitis in paranasal sinuses (PNS) X-ray; (3) patients with a score above daytime cough symptom score ≥ 2 points and nighttime cough symptom score ≥ 1 point; and (4) patients who voluntarily agreed to participate in this clinical trial in written form.

The exclusion criteria are as follows: (1) patients with abnormal findings that could cause cough on the result of chest X-ray; (2) patients with abnormal findings in pulmonary function test (forced expiratory volume (FEV) <80% predicted or forced expiratory volume in 1 s (FEV1)/forced vital capacity (FVC) <70%); (3) patients diagnosed with acute respiratory disease, including upper airway tract within 1 month; (4) patients diagnosed with chronic respiratory disease (chronic obstructive pulmonary disease, bronchial asthma, bronchiectasis, interstitial lung disease, or other chronic respiratory diseases) within 2 years; (5) patients diagnosed with gastroesophageal reflux disease within 1 year; (6) patients who take steroids, anti-leukotrienes, anticholinergics, long-acting β2 agonists, antihistamines, antitussives, and expectorants within 2 weeks before participation in this clinical trial; (7) patients who take proton pump inhibitors, histamine receptor antagonists, mucosal protective drugs, gastrointestinal motility promoters, antacids, antidepressants, anxiolytics, and lower esophageal sphincter agonists for treatment of gastroesophageal reflux disease symptoms within 2 weeks before participation in the clinical trial; (8) patients who have been taking or have taken angiotensin-converting-enzyme inhibitors within 4 months; (9) patients with liver or renal impairment (alanine aminotransferase, aspartate aminotransferase, alkaline phosphatase, and creatinine ≥ 2 times the upper normal limit at screening); (10) patients with generic disorders, such as galactose intolerance, Lapp lactase deficiency, or glucose-galactose malabsorption; (11) patients with comorbidities that interrupt the treatment of cancers or clinical significant disorders of the kidney, liver, psychiatric system, cardiovascular system, respiratory system, endocrine system, and central nervous system, which interrupt assessments of efficacy and safety or completion of the clinical study; (12) patients with unregulated hypertension (systolic blood pressure 160 mmHg ≥ or diastolic blood pressure 100 mmHg ≥ in relaxation period for 10 min); (13) patients with unregulated diabetes mellitus (fasting blood sugar ≥ 180 mg/dL); (14) patients with history of hypersensitivity reaction to materials of drugs used in this clinical trial; (15) patients with a history of alcoholism or substance abuse; (16) current smokers or patients with history of smoking more than 30 pack-year; (17) pregnant or lactating women; (18) patients who do not use medically acceptable contraception (e.g., intrauterine device with proven pregnancy failure rate in spouse or partner, simultaneous use of barrier method for men or women with spermicide, or surgical procedures, such as vasectomy, tubectomy, tubal ligation, or hysterectomy for oneself or one’s partner) during the period from obtaining consent to completion of participation in this trial; (19) patients who participated in other clinical trials within 30 days before participation in this clinical trial; and (20) patients determined by investigators to be ineligible to participate in this trial.

#### 2.2.2. Sample Size

With reference to an exploratory pilot study in which a sample size of 12 per group was recommended [26], the sample size of this trial was 24 participants (12 per group). This is the first randomized controlled trial to investigate the efficacy, safety, and economic evaluation of WBGCT for UACS-induced chronic cough [18]. Considering a 20% dropout rate, 30 participants (15 per group) will be recruited.

#### 2.2.3. Recruitment

Participants will be recruited from regular promotions through mass media, such as newspaper leaflets, daily newspapers, and advertising posters and brochures displayed in the hospital’s outpatient departments. When the recruitment of participants is delayed during the clinical trial, local advertisements will be displayed (subway, bus, and apartment notice boards). Cooperation with patients with chronic cough from online communities, local groups, or societies will also be considered.

### 2.3. Interventions

#### 2.3.1. WBGCT Group

Participants in the WBGCT group will be instructed to take three packs of the investigational medicine (WBGCT) before or between meals three times per day for 2 weeks. WBGCT consists of six medicinal herbs: Ephedrae Herba 3.75 g, Zizyphi Fructus 0.67 g, Gypsum Fibrosum 1.25 g, Atractylodis Rhizoma 2.50 g, Glycyrrhizae Radix et Rhizoma 1.25 g, and Zingiberis Rhizoma Recens 0.83 g, which is manufactured as light brown granules. WBGCT will be purchased from the Kyungbang Pharm Corporation (Incheon, Republic of Korea), a company that has obtained authorization from the Korea Good Manufacturing Practice.

#### 2.3.2. Control Group

Participants in the control group will be instructed to take three packs of active comparator (SCRT) before or between meals three times per day for 2 weeks. SCRT consists of eight medicinal herbs: Ephedrae Herba 0.3 g, Paeoniae Radix 0.51 g, Pinelliae Tuber 0.47 g, Schisandrae Fructus 0.55 g, Cinnamomi Ramulus 0.04 g, Asiasari Radix et Rhizoma 0.18 g, Glycyrrhizae Radix et Rhizoma 0.34 g, and Zingiberis Rhizoma 0.28 g, which is manufactured as brown granules. SCRT will be purchased from the Kyungbang Pharm Corporation. The ingredients and compositions of the WBGCT and SCRT are listed in Table 2.

### 2.4. Outcome Measures

#### 2.4.1. Primary Outcome Measures: Cough Symptom Score (CSS)

The CSS is a rating scale for cough frequency during the day and at night. The score for each question ranges from 0 to 5 (with the total score ranging from 0 to 10) [27]. CSS has been used to assess the severity of cough and the efficacy of therapy in many studies on cough [28,29]. In this study, the Korean version of the CSS will be used, which has proven its validity and clinical utility, as well as acceptable repeatability and responsiveness to changes in patients with chronic cough [30]. Participants will evaluate their cough symptoms twice a day using a cough diary (assessment of daytime CCS from 08:00 to 20:00 at 8 PM and assessment of nighttime CSS from 20:00 to 08:00 at 8 AM). CSS will be assessed at visits 1, 2, 3, and 4.

#### 2.4.2. Secondary Outcome Measures

(1)Cough Visual Analog Scale (VAS).

The VAS evaluates cough severity and frequency. The score ranges from 0 to 10 (0 means “no cough” and 10 means “cough as severe as it could possibly be”) [31]. The VAS will be assessed on visits 2, 3, and 4.

(2)Nasal Discharge Score (NDS)

The NDS assesses the severity of two symptoms: nasal discharge (anterior nasal discharge score) and mucous globus sensation in the back of the throat (posterior nasal discharge score), ranging from 0 to 3 (0 = none; 1 = mild; 2 = moderate; 3 = severe) [6]. Since the symptoms of UACS include nasal symptoms, such as nasal discharge and rhinorrhea, symptoms related to nasal discharge will be assessed using the NDS in this trial [6]. The NDS will be evaluated on visits 2, 3, and 4.

(3)Questionnaire of Clinical Symptoms of Cough and Sputum (QCSCS)

The QCSCS is developed from the Clinical Asthma Measurement Scale in Oriental Medicine-V (CAMSOM-V) in the Traditional Korean Medicine Clinical Practice Guidelines for Antitussives and Expectorants, remodeled for cough and sputum [32,33]. The items for the QCSCS are as follows: (1) cough (frequency, intensity, and sensitivity), (2) sputum (frequency, volume, difficulty in coughing, appearance, and color), (3) activities of daily living, and (4) nighttime sleeping. Each item is measured using a 4-point scale, with a maximum total score of 40 points. The QCSCS will be assessed on visits 2, 3, and 4.

(4)Leicester Cough Questionnaire-Korean version (LCQ-K)

The LCQ-K evaluates the quality of life of patients with a cough and consists of 19 items divided into three parts: psychological, social, and physical aspects. Each item ranges from 1 to 7, with higher scores indicating a better quality of life. Previous studies have shown a correlation between the LCQ score and the severity of cough symptoms. The LCQ-K has been translated into Korean and verified for its validity and reliability [34,35]. The LCQ-K will be assessed on visits 2, 3, and 4.

(5)Integrative Medicine Outcome Scale (IMOS) & Integrative Medicine Patient Satisfaction Scale (IMPSS)

The IMOS evaluates symptom improvement. In this trial, the IMOS will be assessed by the investigators. The IMPSS is a rating scale used to assess treatment satisfaction of patients [36]. The scores on both scales range from 1 to 5. The IMOS and IMPSS will be evaluated on visits 3 and 4.

(6)5-level EuroQol 5-Dimensional Questionnaire (EQ-5D-5L)

The EQ-5D-5L is a self-rating questionnaire that evaluates the quality of life of the general population or patients with various diseases and consist of five dimensions (mobility, self-care, usual activities, pain/discomfort, and anxiety/depression), each with five levels. In addition, participants rate their health by marking a vertical VAS labeled number from 0 to 100 (100 means “the best health you can imagine” and 0 means “the worst health you can imagine”) [37]. The EQ-5D-5L will be evaluated on visits 2, 3, and 4.

#### 2.4.3. Exploratory Outcome Measures: Pattern Identification for Chronic Cough Questionnaire (PICCQ)

The PICCQ is used to identify patterns of chronic cough in each participant and is conducted at baseline of the trial. Chronic cough is classified into five patterns according to its signs and symptoms: cold wind, phlegm turbidity, fire-heat, lung deficiency, and kidney yang deficiency [38]. These pattern identifications are important diagnostic criteria used to determine appropriate medications and other therapies in Korean Medicine. Thus, we will investigate the correlation between pattern identification and the effectiveness of WBGST and SCRT.

### 2.5. Safety Assessment

For the safety assessment, adverse events (AEs), vital signs, electrocardiograms, and clinical laboratory tests (complete blood count, liver function test, and urine analysis) will be assessed. AEs are harmful and unintended signs, symptoms, and diseases in patients taking investigational drugs, which do not necessarily imply a causal relationship with the investigational drugs. Vital signs and AEs will be assessed at every visit after the investigational medications are administered (visits 2, 3, and 4). The investigators will record the date of onset, severity, results of AEs, action taken against investigational medications, causal relationship between AEs and investigational medications, other suspected medicines, and treatments for AEs. When AEs occur, the principal investigator should conduct continuous follow-up until they disappear, or the patients are in a stable condition. All serious AEs will be reported to the principal investigator within 24 h.

### 2.6. Economic Evaluation

The primary economic endpoint is the incremental cost-utility ratio (ICUR) calculated at every visit using the EQ-5D-5L. The secondary economic endpoint is the incremental cost-effectiveness ratio (ICER) calculated at every visit using the variables for clinical effectiveness. A cost evaluation will be performed using micro-costing based on data, including survey results, medical records, and administration data, while gross costing will be conducted as needed. Cost assessments will be performed using surveys conducted on visits 2, 3, and 4. For participants who dropped out of the trial early, the costs until the investigational medication ended will be collected.

### 2.7. Assignment of Interventions

#### 2.7.1. Allocation

The participants are allocated according to blocked randomization without stratification. A randomized allocation sequence will be assigned to each group based on a computer-generated random number table created by an independent statistician using SAS version 9.4 statistical software (SAS Institute. Inc., Cary, NC, USA). After consent to the clinical trial, the participants’ identification code will be assigned to those who meet the inclusion and exclusion criteria. The participants will be allocated to the WBGCT and control groups at an allocation ratio of 1:1. The randomization table will be retained separately by the independent statistician to maintain blinding. The assignment of participants will be conducted according to the assignment table generated by a random assignment method that can be specifically planned and reproduced in advance. According to the randomized assignment table, the participants’ identification codes will be kept in an opaque envelope and a cabinet with double locks.

#### 2.7.2. Blinding

As this clinical trial is double-blinded, the allocation of participants will be blinded to both the participants and investigators until the clinical trial is completed. The investigators will not be involved in the randomization and do not know what type of drugs the participants will receive. WBGCT and SCRT will be packed in the same wrapping to blind the participants. The identification codes will be developed to prevent the participants from distinguishing their assigned group. Participants will be assigned random numbers and receive medication packages corresponding to the identification code. The investigators will manage the allocation of identification codes, which should not be disclosed until the clinical trial is completed, except in cases of emergency in which the identification code needs to be identified.

### 2.8. Data Management and Monitoring

The investigators should be acquainted with the trial protocol and execute the trial according to the protocol and related regulations. Data will be collected during each visit. The investigator should preserve documents related to the clinical trial during the 3 years after the end of the trial.

Monitoring will be conducted through regular visits and occasional telephone calls by the clinical research associates (CRAs). The CRA will determine whether the clinical trial is being performed in compliance with the protocol, whether AEs are being reported and recorded on case report forms (CRFs), and inspect the consent of the participants and management of investigational medications. The monitors who conduct the monitoring will check the progress of the clinical trials, and problems will be consulted with the investigators.

### 2.9. Statistical Analysis

In this trial, an efficiency analysis will be mainly conducted using a full analysis set (FAS) analysis, and a per-protocol set (PPS) analysis will be performed as needed. The FAS group is defined as the analysis group based on intention-to-treat (ITT) principles. Participants in the FAS group receive the intervention at least once when evaluating efficiency. All data will be included in the analysis after excluding participants who do not meet the exclusion criteria (those who violate the inclusion/exclusion criteria, have never taken investigational medications, or do not provide any data after screening). The PPS analysis group consists of participants who complete the entire trial course with ≥70% compliance to investigational medications without violating the protocol.

Descriptive statistics for each group regarding demographic characteristics and baseline data will be presented. Continuous variables will be expressed as means and confidence intervals, and an independent *t*-test or Wilcoxon rank-sum test will be performed depending on their normality. Categorical variables are expressed as frequencies and percentages, and χ2 test or Fisher’s exact test will be conducted. The primary outcome measures will be analyzed using the mixed effect model repeated measure (MMRM), setting each group and visit as a fixed effect and participants as a random effect. If necessary, the fixed effects will include variables with significant differences in demographic characteristics or variables that can affect chronic cough. Analysis of the primary outcome measures are expressed as mean and 95% confidence interval (CI). Among the secondary outcome measures, CSS scores for 4 weeks against baseline and cough VAS, NDS, LCQ-K, and QCSCS scores will be analyzed using the same methods as the primary outcome measure. Depending on normality, Student’s paired *t*-test or Wilcoxon signed-rank test will be used to analyze differences in scores before and after treatment. The IMOS and IMPSS scores at 2- and 4-week intervals will be analyzed using Fisher’s exact test. Repeated-measures analysis of variance (RM ANOVA) will be used to compare the differences in the changes in CSS score trends between the groups, and Dunnett’s procedure for multiple comparison correction will be used. When performing the Student’s paired *t*-test, Wilcoxon signed-rank test, or RM ANOVA, a missing data will be substituted with Last Observation Carried Forward (LOCF) or Multiple Imputation (MI).

A safety evaluation will be conducted on participants who receive Interventions at least once. The number of AEs between the groups will be compared using Fisher’s exact test. Differences in laboratory test variables before and after treatment will be analyzed using the Student’s paired *t*-test or Wilcoxon signed-rank test.

Economic evaluations will be performed from both social and limited social perspectives. In sensitivity analysis, the available variables will be analyzed using a one-way sensitivity analysis with a tornado diagram. If necessary, a probabilistic sensitivity analysis will be conducted using the distribution and representative values of the variables.

## 3. Ethics and Dissemination

The clinical trial was performed in accordance with the Declaration of Helsinki, Korean Good Clinical Practice (GCP) Guidelines, related laws, and protocols. This study was approved by the Institutional Review Board (IRB) of Daejeon University Daejeon Korean Medicine Hospital (approval number: DJDSKH-22-DR-15) in September 2022. This study was registered in October 2022 at the National Clinical Trial Registry Clinical Research Information Service (https://cris.nih.go.kr (accessed on 20 August 2023)) with the identifier number KCT0007822.

Before the clinical trial, the investigators will provide all information related to the trial through written informed consent forms. Participants should be allowed sufficient time to obtain their approval. After the participants voluntarily decided to participate in the trial, they will sign a document containing all the instructions. All identifiable patient data will be kept confidential in a controlled-access laboratory archive. In all documents relevant to the clinical trial, including the CRFs, the participants should be recorded and differentiated by their identification code and initial. Monitors and inspectors can access the records of the participants to monitor, inspect, and manage the progress of the trial.

## 4. Discussion

UACS is a common cause of chronic cough characterized by nasal symptoms and throat discomfort, which can deteriorate the patients’ physical and mental health [7]. As there is no specific diagnosis for UACS, physicians provide treatment in accordance with the suspected cause based on symptoms, physical examinations, and radiography. When it is difficult to determine the cause of symptoms, empiric medications such as first-generation antihistamines and decongestants are usually used for UACS. However, using antihistamines in patients with UASC may induce sedation, fatigue, memory loss, and decreased concentration, making the long-term use of these medications difficult for patients [39,40]. Therefore, alternative medicines that are effective and safe against UASC-induced chronic cough are required.

WBGCT is an herbal medication used to treat various diseases, including rhinitis and allergic diseases. In previous studies, WBGCT has antioxidant and anti-inflammatory effects by suppressing nitric oxide synthase, cyclooxygenase-2, cytokines, and interleukin-4 (IL-4) [18,41], which may affect UACS-induced chronic cough related to airway inflammation and allergic reactions [42]. Other studies have reported that WBGCT has clinical effects on respiratory diseases, such as allergic rhinitis and bronchial asthma [15,16].

In recent studies, the phytochemical and pharmacological mechanisms of the ingredients and active compounds in WBGCT have demonstrated potential effects that alleviate symptoms of UACS-induced chronic cough. Ephedrae Herba has been widely used for treating respiratory diseases such as asthma, allergic rhinitis, and common cold [21]. Ephedrine, a major active compound in Ephedrae Herba, has inhibitory effects on the infiltration of inflammatory cells into pulmonary tissues and the secretion of inflammatory mediators, depressing levels of tumor necrosis factor α (TNF-α) and interleukins. It is also a bronchodilator, stimulating β-adrenergic receptors in the lung [43,44,45]. Atractylodis Rhizoma has been used for treating dyspepsia, viral infection, and rheumatic diseases [46]. Among its compounds, atractylon has anti-inflammatory effects and alleviates influenza virus-induced pulmonary injury by decreasing pro-inflammatory cytokines (IL-1β, IL-6, and TNF-α) [47]. Gypsum Fibrosum, a mineral composed of calcium sulfate, has antipyretic and anti-inflammatory effects, which are achieved by reducing secretion of IL-4 in mice splenocytes [48,49]. Glycyrrhizin from Glycyrrhizae Radix et Rhizoma alleviates inflammation and oxidation by depressing the expression of cytokines and reactive oxygen species (ROS), inhibiting virus replication as well [50]. Gingerol, a phenolic compound of Zingiberis Rhizoma Recens, has antibacterial and antioxidative effects from oxidative damage [51]. These results support the therapeutic effects of WBGCT in USAC-induced chronic cough. The potential mechanisms of therapeutic effects on UACS-induced chronic cough ingredients of WBGCT are shown in Figure 2.

SCRT, the investigational medication in the active control group, is an herbal medication used for respiratory diseases, such as allergic rhinitis and the common cold [24,52], which has inhibitory effects on the infiltration of inflammatory cells in the airway [22]. Due to the therapeutic effects of SCRT on respiratory diseases, SCRT is selected for comparison with WBGCT. In this study, we also observe the clinical effects of SCRT on UACS-induced chronic cough, as no other studies on UACS using SCRT have been conducted.

This study has some limitations. There is no control group receiving a placebo; therefore, it is impossible to compare the effects of WBGCT with a placebo. We will compensate for this restriction with reference to minimal clinically important differences in each outcome to observe the clinical effectiveness of the WBGCT and plan to add a placebo group in the next confirmatory trial. Moreover, our study has a small sample size, as it is a preliminary trial designed to assess the feasibility of the study protocol. After obtaining the effect size based on our study results, we will calculate the appropriate sample size needed to confirm the efficacy of WBGCT on UACS in the next trial. Lastly, due to the short period of the study, it is difficult to assess the long-term effects of WBGCT. Considering that we target chronic cough induced by UACS, further study with longer period to demonstrate long-term effects will be needed.

## 5. Conclusions

This study aims to demonstrate the effects of WBGCT on UACS-induced chronic cough. As WBGCT has been used widely for respiratory diseases over a long period of time with few side effects, WBGCT is expected to be a safe and effective agent for UACS-induced chronic cough. Moreover, considering that our intervention is Korean medicine, we used pattern identification to obtain additional information on whether certain patterns of patients respond better than others to the treatment. Additionally, we perform an economic evaluation of WBGCT compared to SCRT, expecting the expansion of newly applied health insurance-covered herbal formulas. Our study will support the efficacy, safety, and economic feasibility of WBGCT and provide evidence for its clinical use.

## Figures and Tables

**Figure 1 healthcare-11-02733-f001:**
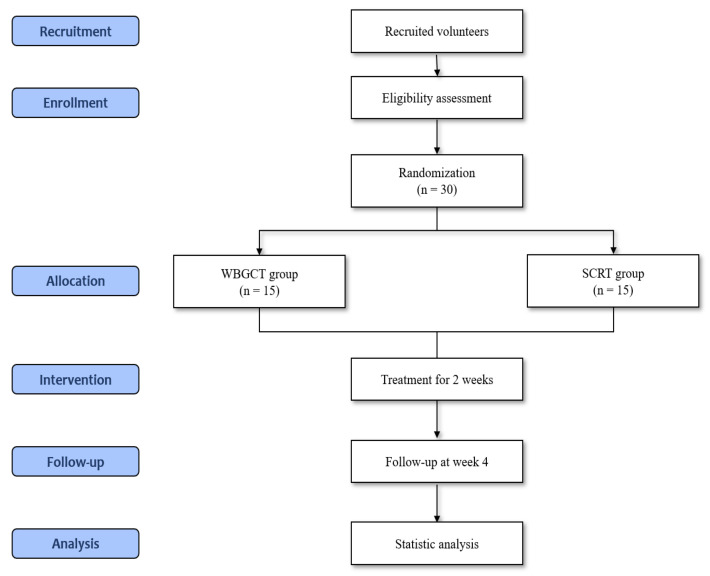
Flowchart of the study plan. SCRT; Socheongryong-tang, WBGCT; Wolbigachul-tang.

**Figure 2 healthcare-11-02733-f002:**
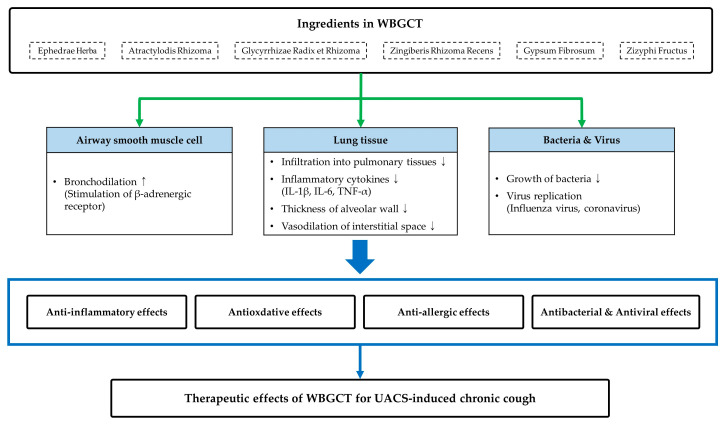
The potential mechanisms of ingredients in WBGCT for UACS-induced chronic cough. Previous studies demonstrated that the ingredients had anti-inflammatory, antioxidative, anti-allergic, antibacterial, and antiviral effects, affecting airway smooth muscle and lung tissue [43,44,45,46,47,48,49,50,51]. UACS; upper airway cough syndrome, WBGCT; Wolbigachul-tang.

**Table 1 healthcare-11-02733-t001:** Schedule of enrollment, interventions and outcome measurements.

	Study Period
	Enrollment	Allocation	Post-Allocation	Follow-Up
Visit	1	2	3	4
Timepoint	−2 Weeks	0	Week 2 ± 4 Days	Week 4 ± 4 Days
Enrollment				
Informed consent	X			
Eligibility screen	X			
Allocation		X		
Demographic characteristics	X			
Social and economic characteristics	X			
Medical history	X			
Chest X-ray, EKG	X			
Vital signs	X	X	X	X
Pulmonary Function Test	X			
Paranasal Sinuses X-ray	X			
Interventions				
WBGCT		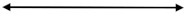	
SCRT		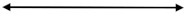	
Assessments				
Clinical laboratory test ^1^	X		X	
Pregnancy Diagnosis Test	X			
CSS	X	X	X	X
Cough VAS		X	X	X
NDS		X	X	X
Cough and sputum severity standard		X	X	X
LCQ-K		X	X	X
IMOS & IMPSS			X	X
EQ-5D-5L		X	X	X
PICCQ	X			
Adverse reaction assessment		X	X	X
Compliance test		X	X	

^1^ CBC (erythrocyte count, leukocyte count, differential count, hemoglobin, hematocrit, platelet count, and erythrocyte sedimentation rate); LFT (AST, ALT, γ-GTP, ALP, BUN, creatinine, total bilirubin, and glucose); and U/A (urine chemical test paper, and urine microscopy). Abbreviations. ALP, alkaline phosphatase; ALT, alanine aminotransferase; AST, aspartate aminotransferase; BUN, blood urea nitrogen; CBC, complete blood count; CSS, cough symptom score; EQ-5D-5L, 5-level EuroQol 5-dimensional questionnaire; γ-GTP, gamma-glutamyl transpeptidase; IMOS, integrative medicine outcome scale; IMPSS, integrative medicine patient satisfaction scale; LCQ-K, Leicester cough questionnaire-Korean version; LFT, liver function test; NDS, nasal discharge score; PICCQ, pattern identification for chronic cough questionnaire; SCRT, Socheongryong-tang; U/A, urine analysis; VAS, visual analog scale; WBGCT, Wolbigachul-tang.

**Table 2 healthcare-11-02733-t002:** Composition of Wolbigachul-tang and Socheongryong-tang.

Latin Name	Amount (g)
Wolbigachul-tang	
Ephedrae Herba	3.75
Zizyphi Fructus	0.67
Gypsum Fibrosum	1.25
Atractylodis Rhizoma	2.50
Glycyrrhizae Radix et Rhizoma	1.25
Zingiberis Rhizoma Recens	0.83
Socheongryong-tang	
Ephedrae Herba	0.30
Paeoniae Radix	0.51
Pinelliae Tuber	0.47
Schisandrae Fructus	0.55
Cinnamomi Ramulus	0.04
Asiasari Radix et Rhizoma	0.18
Glycyrrhizae Radix et Rhizoma	0.34
Zingiberis Rhizoma	0.28

## Data Availability

The datasets used or analyzed during the current study will be available from the corresponding author upon reasonable request.

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
