# Peer review of "Efficacy, Safety and Economic Evaluation of Wolbigachul-Tang for Chronic Cough Due to Upper Airway Cough Syndrome (UACS): A Study Protocol for Randomized, Double-Blind, Active-Comparator Controlled, Parallel, Exploratory Clinical Trial"

_healthcare, 2023, doi:10.3390/healthcare11202733_

Round 1

Reviewer 1 Report

I support this clinical trial study and believe it has the potential to significantly contribute to the field of UACS-induced chronic cough research.

1.     2.1 Study design: The study is relatively short (4 weeks), which may need more time to assess the efficacy and safety of WBGCT fully. It needs to address the long-term impact and focus on short-term effects.

2.     2.2.1 Inclusion/Exclusion Criteria: The authors should provide more information on the randomization and blinding procedures. For example, they should specify how they will generate the randomization sequence and ensure the participants and investigators are blinded to the treatment allocation. The criteria (20) can be catch-all but should be used judiciously to prevent subjective bias.

3.     2.7.1 Allocation: While blocked randomization is adequate, the block size should be predetermined and undisclosed to ensure unpredictability. The authors don’t specify the block size in this article; please explain.

4.     2.9 Statistical Analysis: Consider using a power analysis to determine the optimal sample size for the study. This will ensure the study is adequately powered to detect a clinically meaningful difference between the two interventions.

5.     Are there any data in this study that needs to be included? How did the authors treat the missing data?

Reviewer 2 Report

The study protocol entitled "Efficacy, safety and economic evaluation of Wolbigachul-tang for chronic cough due to upper airway cough syndrome (UACS): A study protocol for randomized, double-blind, active comparator controlled, parallel, exploratory clinical trial ” by Woo et al. evaluated the efficacy, safety, and economic feasibility of Wolbigachul-tang (WBGCT), an herbal medication for UACS- induced chronic cough.  In my opinion, the manuscript is need major revision to be suitable to publish in healthcare journal due to the following issues,

1- Please review all manuscript by native speakers, because there are some important typographical accentuation and grammatical errors throughout the paper.

2- There are many references in the introduction section that need to update.

3- The dose used in the treatment protocol from the WBGCT herbal formula must be indicated in the Abstract section

4- Line 70 please delete “this is"

5- Line 84 the number of males and females on whom the study was conducted must be determined separately and not the total number

6- Lines 371-373 the authors mention” WBGCT has been widely used for respiratory diseases for a long period of time with few side effects.  So, some previous studies of WBGCT side effects must be pointed out in the introduction section.

7- Cytotoxicity must be studied on normal lung fibroblast WI-38 cells to determine side effects of formulation WBGCT with the concentrations as used in this protocol.

8- The discussion section is insufficient. It must be rewriting with an explanation of the most important active compounds found in the herbal species used in Formula WBGCT and their role as antioxidants and anti-allergic and anti-respiratory infections.

9- I suggest designing a diagram that explains the mode of action of the most important active ingredients in WBGCT formula onto the cells of the respiratory system (Pathways).

Minor editing of English language

Reviewer 3 Report

This study protocol proposed a study aiming at the investigation of the efficacy, safety, and economic feasibility of Wolbigachul-tang (WBGCT) in treating chronic cough. The protocol provides detailed information about the study design, however there are certain improvements that can be made before publishment. Please see detailed comments below.

1. In Abstract, line 31. Is UACS a typo here, whereas WBGCT should be used? 

2. In Introduction, the authors stated to evaluate the economic feasibility of WBGCT by comparing it with SCRT. It would benefit the readers if the authors can provide more information about the SCRT's efficacy, safety and so on regarding the treatment of UACS-induced chronic cough. It is recommended to include more background information about choosing SCRT for control group.

3. Although the authors mentioned that first-generation antihistamines or decongestants are empirically administered for patients with UACS-induced cough, there is no antihistamines or decongestants studied or used in control group in later study design.

Round 2

Reviewer 2 Report

Thank you for your very kind response, as well as the requested suggestions

Reviewer 3 Report

All my concerns have been addressed in the updated manuscript. Therefore, the publication of the article in current form is recommended.